# Time-Course Transcriptomic Analysis Reveals Molecular Insights into the Inflorescence and Flower Development of *Cardiocrinum giganteum*

**DOI:** 10.3390/plants13050649

**Published:** 2024-02-27

**Authors:** Yu Wei, Aihua Li, Yiran Zhao, Wenqi Li, Zhiyang Dong, Lei Zhang, Yuntao Zhu, Hui Zhang, Yike Gao, Qixiang Zhang

**Affiliations:** 1Beijing Laboratory of Urban and Rural Ecological Environment, School of Landscape Architecture, Beijing Forestry University, Beijing 100083, China; weiyu@chnbg.cn (Y.W.); yiranzhao@bjfu.edu.cn (Y.Z.); yuntao_zhu@bjfu.edu.cn (Y.Z.); 2Key Laboratory of National Forestry and Grassland Administration on Plant Ex Situ Conservation, Beijing Botanical Garden, Beijing 100093, China; liaihua@chnbg.cn (A.L.); liwenqi@chnbg.cn (W.L.); dongzhiyang@chnbg.cn (Z.D.); zhanglei@chnbg.cn (L.Z.); zhanghui@chnbg.cn (H.Z.)

**Keywords:** *Cardiocrinum giganteum*, transcriptome, inflorescence, flower development, synchronization

## Abstract

*Cardiocrinum giganteum* is an endemic species of east Asia which is famous for its showy inflorescence and medicinal bulbs. Its inflorescence is a determinate raceme and the flowers bloom synchronously. Morphological observation and time-course transcriptomic analysis were combined to study the process of inflorescence and flower development of *C*. *giganteum*. The results show that the autonomic pathway, GA pathway, and the vernalization pathway are involved in the flower formation pathway of *C*. *giganteum*. A varied ABCDE flowering model was deduced from the main development process. Moreover, it was found that the flowers in different parts of the raceme in *C*. *giganteum* gradually synchronized during development, which is highly important for both evolution and ecology. The results obtained in this work improve our understanding of the process and mechanism of inflorescence and flower development and could be useful for the flowering period regulation and breeding of *C*. *giganteum*.

## 1. Introduction

Giant lily [*Cardiocrinum giganteum* (Wall.) Makino] is a perennial bulbous plant of the family Liliaceae that has showy lily-like flowers. This species is endemic to East Asia and distributed mainly in temperate and subtropical woodlands in southwest China [1]. Under natural conditions, the developmental period of transition from the vegetative to the reproductive phase is long, but the flowering period is short, which limits garden application. Therefore, it is necessary to explore the flowering characteristics and underlying mechanisms of this plant to prolong its flowering period. The presence or absence of a terminal flower contributes to a determinate or indeterminate inflorescence [2]. The inflorescence of *C. giganteum* terminates with not a flower, but instead a twisted bract-like structure, thus limiting the number of flowers on the inflorescence and making it a determinate raceme. Thus, *C. giganteum* is considered to be a potentially suitable species for studying inflorescence and flower development in plants with determinate inflorescence, especially those that terminate with a bract-like structure.

Flowers are unique organs of angiosperms with remarkable diversity and complexity. Flower development can be subdivided into several major procedures, such as floral induction, floral meristem formation, and floral organ development and maturation [3]. Floral induction refers to the process of plant transformation from vegetative growth to reproductive growth. In response to genetic factors and various environmental signals, the shoot apical meristem (SAM) is converted into the inflorescence meristem (IM). According to genetic studies of *Arabidopsis* and crops, more than 300 genes participate in complex signaling pathways involved in floral induction, including the age, autonomic, circadian clock, and gibberellin (GA) pathways, which respond to intracellular and intercellular signals, as well as the vernalization and photoperiod pathways, which respond to environmental factors [4].

During the flower meristem formation stage, auxin accumulates in the IM area through polar transport, and the IM is further transformed into the flower meristem (FM) and flower primordium [5]. The *PIN-FORMED* (*PIN*) and *YUCCA* (*YUC*) genes are the most important factors that control the biosynthesis and polar transport of auxin. The quantity and structure of floral organs are considerably affected by the loss of these genes [6]. Auxin triggers flower initiation through the *AUXIN RESPONSE FACTOR* (*ARF*) *MONOPTEROS* (*MP*), which activates *AINTEGUMENTA* (*ANT*), *AINTEGUMENTA-LIKE 6* (*AIL6*) [7], *ARABIDOPSIS HISTIDINE PHOSPHOTRANSFER PROTEIN 6* (*AHP6*), and other genes to promote flower growth [8,9]. In the model plant *Arabidopsis thaliana*, *FLOWERING LOCUS T* (*FT*) and *SUPPRESSOR OF OVEREXPRESSION OF CONSTANS 1* (*SOC1*) can integrate many flowering genes regulated by multiple pathways and subsequently activate downstream FM identity genes, such as *LEAFY* (*LFY*) and *APETALA1* (*AP1*), to cooperate in promoting flowering [10]. Many transcription factors regulate *FT* by directly binding to the *FT* promoter [11]. The initiation and formation of FM are generally believed to be related to the activities of stem cells, which are regulated by *WUSCHEL* (*WUS*) and *CLAVATA3* (*CLV3*) [12,13,14]. Current research shows that *TERMINAL FLOWER 1* (*TFL1*) in *Arabidopsis thaliana* and *CENTRORADIALIS* (*CEN*) in *Antirrhinum majus* are expressed specifically in the IM and depress the expression of the key flowering genes *LFY* and *FLORICAULA* (*FLO*) so that the SAM maintains meristem characteristics [15,16]. IM identity and/or indeterminacy may be regulated by the mutual repression of *TFL1*/*APETALA2* (*AP2*) and *AP1*/*FRUITFUL*(*FUL*)/*AGAMOUS-LIKE 79* (*AGL79*) [17]. After the formation of the IM, the FM develops from the flank of the inflorescence primordium, which eventually promotes the formation of floral organs [18].

Floral organ development is a process in which a tiny primordium develops into a full-sized floral organ. The floral organs of angiosperms are usually composed of sepals, petals, stamens and carpels, and each organ has its own developmental program. The development of each flower involves several sub-programs, which are expressed under the control of regulatory genes. The ABC model was proposed based on the study of mutants of *A. thaliana* and *A. majus*. The four-whorled structure of flowers is determined by the combined expression of three types of genes, A, B and C, in which A specifies the development of the sepal, A + B specifies the development of the petal, B + C specifies the development of stamen, and C specifies the development of carpel; A and C are antagonistic [19,20]. Subsequent genetic analyses in *Arabidopsis* revealed five distinct functional genes, including two genes mediating the A function, *AP1* [21] and *AP2* [22]; two genes mediating the B function, *APETALA 3* (*AP3*) [23] and *PISTILLATA* (*PI*) [24]; and one gene for the C function, *AGAMOUS* (*AG*) [25], which encodes transcription factor that is directly or indirectly involved in the formation of floral organs. Later, the model was modified to the ABCDE model through extensive research on the mutant of a floral organ identity defect. *FLORAL* BINDING PROTEIN *11* (*FBP11*), which is related to ovule development, was named the D gene, while the *SEPALLATA 1*/*2*/*3* (*SEP1*/*2*/*3*) gene, which is indispensable for the whole development of floral organs, was named the E gene. Together, these five genes provide sufficient conditions for the development of flower organs [26,27,28]. 

Floral organ maturation is the process through which flowers acquire their final characteristics, such as color and flavor. This process is very important for flowers, because these characteristics are crucial to reproduction and defense. The color of floral organs depends on the type, content and spatial arrangement of pigments [29]. Anthocyanins have been identified as the main pigments that give petals a set of colors [30]. The synthesis pathways of anthocyanins and important related genes, such as *CHALCONE SYNTHASE* (*CHS*), *FLAVANONE 3-HYDROXYLASE* (*F3H*), *ANTHOCYANIDIN SYNTHASE* (*ANS*), *DIHYDROFLAVONOL 4-REDUCTASE* (*DFR*) and *MYB* genes, have been fully studied and characterized [31,32]. The taste of the plant is related to the type and concentration of the metabolites that accumulated in cells [33]. Similar to pollen and nectary, sweetness mainly depends on the sugars produced in floral organs [29]. In the flower development process, anther development deserves special attention, because the release of pollen is critical for plant reproduction. The development of anthers includes early organ differentiation, meiosis, tapetum development, microspore maturation, and apoptosis. Many genes related to anther development, such as *WUS*, *AG*, *AP3*, *PI*, *SPOROCYTELESS* (*SPL*), *JAGGED* (*JAG*) and *NUBBIN* (*NAB*), have been cloned [34,35,36]. 

All the above-mentioned regulatory factors are based mainly on the study of model species, but the conservation, convergence, and differences in homologous genes between model and non-model species during inflorescence and flower development are still unclear. In addition, research on *C*. *giganteum* has focused mainly on bulb development [37], seed dormancy [38], and phylogenetic analysis [1,39]. Pollen development during the process of inflorescence differentiation in this species has been studied, but the morphological and molecular mechanisms related to other developmental stages have not been reported. By observing the morphological traits of inflorescence and flower bud, and analyzing the development process of outdoor plants in Beijing, this study identified key genes involved in different developmental stages of inflorescence through time-course transcriptomic analysis, and analyzed the changes in key genes involved in flower induction, flower development, anthocyanin synthesis related to flower color presentation and anther development in *C*. *giganteum*. The results obtained in this work improve our understanding of the process and mechanism of inflorescence and flower development, which could be useful for breeding and regulating the flowering period of *C*. *giganteum*.

## 2. Results

### 2.1. Phenology of Inflorescence Development of C. giganteum

According to the morphological analyses, inflorescences development in *C*. *giganteum* can be divided into five stages (Figure 1). During the first developmental stage (S1), the floret primordia gradually differentiated. The floret primordia in the lower part of the inflorescence differentiated first, then in the middle and finally in the upper part. The interval between the appearance of the first and last flower primordia was approximately three months (from early December to early March). At the S2 stage, the floret primordia further differentiated into flower buds. The perianths, stamens and carpels were visible. The pollen-sacs were visible, but no filaments or pollen was visible inside. The scape was elongated considerably at stage S3, and the perianths were stained yellowish and lightly reddish brown. The anthers and filaments of the stamen were clearly separated, but still no pollen was observed inside the anthers. The stigma and the carpel of the pistil were formed. At stage S4, the flower buds spread, and the perianths were nearly the same size as those of the open flower, and were heavily stained reddish brown in the middle of the perianth. The flower opened and scented at stage S5, the anther dehisced, and the pollens were shed. The nectary appeared at the base of the flower.

### 2.2. Transcriptomic and Correlation Analysis of the Five Developmental Stages and Three Inflorescence Parts of C. giganteum

To understand how the inflorescence and flower of *C*. *giganteum* were formed throughout development, we conducted a detailed time-course transcriptomic analysis of the five developmental stages and three inflorescence parts. The transcriptome sequences of all 39 samples were completed, with Q30 reaching 85% (Appendix A). A total of 322,089 circular consensus sequences (CCSs) were obtained, including 272,167 full-length read non-chimeric (FLNC) sequences. Clustering the FLNC sequences resulted in a total of 96,731 consistent sequences and 96,726 high-quality consistent sequences. Redundancy analysis of high-quality consistent sequences resulted in 53,871 transcript sequences. Functional annotation information for 44,454 transcripts was obtained, and detailed annotation information against the different databases is shown in Table 1. The repeatability of replicates of each sample was high (coefficient of determination *R*^2^ > 0.80; Figure 2A); therefore, the expression level of each sample is represented by the average FPKM value of three replicates. Finally, we obtained 44,000 gene expression data points in which genes were expressed in at least one sample (RPKM ≥ 1.0) (Appendix A).

Principal component analysis (PCA) revealed that in S1 and S5 stages, the three parts of the inflorescence were clustered together, but were far from the adjacent developmental stages, indicating that the gene expression patterns of the three parts in the same developmental stage were largely consistent (Figure 2B). The expression patterns of the three parts in the S2 stage were relatively scattered, which indicated that the gene expression patterns among the different parts of the inflorescence were the most different at that stage. The S3 and S4 stage samples were very close. Additionally, the gene expression patterns of the samples in the upper and middle inflorescence parts in the S3 stage were basically combined, indicating that the gene expression patterns of the upper and middle flower buds in the S3 stage were basically the same. We also found that the development of basal flower buds on the inflorescence in the S4 stage were strongly delayed, which was similar to the development process of the middle and upper parts in the S3 stage. The results of the gene expression model based on PCA analysis were basically consistent with the morphological observations (Figure 1B). Similar phenomena were observed for the Pearson correlation coefficient (PCC) and the Spearman’s rank coefficient of correlation (SCC) (Figure 2C). The correlation coefficient values of adjacent development stages were greater than those of non-adjacent development stages. In addition, the correlation coefficient between adjacent parts of the inflorescence was generally greater than that between non-adjacent parts. Interestingly, the PCC and SCC between Lower4 and Upper3 (0.41, 0.56) and Middle3 (0.42, 0.54) were relatively greater than those between the other samples at S3 and S4. This also demonstrated that the development of lower buds at S4 was delayed, and that the development of these buds was similar to that of Upper3 and Middle3. In summary, our study suggested that the different parts of the inflorescence are actually not very different from each other at the early stage of inflorescence development (S1) in terms of the genes expressed. However, during development, the difference in gene expression among different parts increased in the S2 stage, then decreased in the S3 and S4 stages, and was almost the same in the S5 stage. 

### 2.3. Co-Expression Modules and Expression Trends at Different Developmental Stages of the Inflorescence

To explore the dynamic changes in gene expression patterns during the inflorescence development process, a weighted gene co-expression network analysis (WGCNA) was performed for genes with a coefficient of variation (CV) greater than 0.5 across the five developmental stages. A total of 38,499, 38,873 and 38,129 genes were used to analyze the upper, middle and lower parts of the inflorescence, respectively, and 12 co-expression modules were identified (Figure 3A,C,E). The number of expressed genes per module ranged from 1922 (Upper, M10) to 5324 (Upper, M12) (Figure 3D,F; Appendix A). The correlation coefficient analysis matrix (Figure 3A,C,E) showed that the genes of modules M1–M4 were highly expressed in the S1 stage, and the genes of modules M4–M6, M6–M8, M8–M10, and M10–M12 were highly expressed mainly in the S2, S3, S4, and S5 stages, respectively. Furthermore, the six modules M1, M5, M7, M9, M11, and M12 were strongly correlated with the five developmental stages S1, S2, S3, S4, and S5 (PCC ≥ 0.85; PCC < 0.30 in the other stage), while the other modules were related to two or more adjacent stages (PCC ≥ 0.30). Trend analysis of gene expression (Figure 3B,D,F) revealed that M1 module genes highly correlated with the S1 stage were highly expressed only in the S1 stage and were poorly expressed in the other stages (S2–S5); moreover, M5, M7, M9, and M12 module genes were highly expressed only in the corresponding S2, S3, S4, and S5 stages but poorly expressed in the other four stages. Only the genes of the M11 module were strongly correlated with the S4 and S5 stages. The genes of these modules might be closely related to the development of inflorescences and flowers at the corresponding stages.

### 2.4. Genes Specifically Expressed during Inflorescence and Flower Development

To investigate the gene functions related to the key developmental stages of the inflorescence, Gene Ontology (GO) enrichment analysis was carried out for genes of 12 modules whose genes were specifically expressed in the five developmental stages. The results for the lower part of the inflorescence (Figure 4A, Appendix A) showed that the representative GO terms “inflorescence morphogenesis”, “floral organ development”, “polarity specification of adaxial/abaxial axis”, “maintenance of inflorescence meristem identity”, “vegetative to reproductive phase transition of meristem” and “specification of petal number” were enriched in M1. Because M1 was strongly correlated with the S1 stage (PCC = 0.99; Figure 3E), the aforementioned GO categories demonstrated that inflorescence development and flower formation occurred at the S1 stage. Terms such as “plant-type secondary cell wall biogenesis”, “response to cytokinin”, “stamen filament development”, “jasmonic acid biosynthetic process” and “brassinosteroid biosynthetic process” were enriched in M5. Moreover, the terms “shoot system morphogenesis”, “anther development”, “flower development”, and “terpenoid biosynthetic process” were enriched in M7. Based on the high correlation coefficient of M5 with S2 (PCC = 1; Figure 3E) and M7 with S3 (PCC = 1; Figure 3E), the enriched GO terms in M5 and M7 suggested that flower development, especially anther development, occurred at stages S2 and S3. GO terms related to flower color, including “flavonoid biosynthetic process”, “terpenoid biosynthetic process”, “positive regulation of anthocyanin metabolic process”, and “monoterpene biosynthetic process”, were enriched mainly in M11 and M12, which were strongly correlated with the S5 stage (PCC = 0.92 and 1, respectively; Figure 3E). The GO term related to pollen development, “pollen sperm cell differentiation”, was enriched in M11. There were also several representative GO terms enriched in other modules, such as “flower development” in M2; “plant ovule development” in M3 and M12; “pollen sperm cell differentiation” in M7; “anther development” in M5, M7 and M8; and “flavonoid biosynthetic process” in M5 and M8.

We further screened the genes related to the development of inflorescence and flowers in the lower part of the inflorescence (Figure 4B). Most of these genes are orthologues of known functional genes in *Arabidopsis*. In stage S1, several orthologues are involved in the transformation from vegetative meristem to reproductive meristem, including *CgLFY* [40], and *CgSPL9* [41]. There are orthologues of *ANT* [42], *KNOTTED1-LIKE HOMEOBOX GENE 6* (*KNAT6*) [43] and *SHOOTLESS* (*STM*) [44], which play key roles in the initiation and maintenance of the floral meristem. In the S2 stage, there are orthologue sequences of *HOMEOBOX PROTEIN 21* (*HB21*) and *PHYTOENE DESATURASE* (*PDS*) [45], which participate in polarity determination, floral organ development and anthocyanin biosynthesis, respectively; homologous genes of *ABORTED MICROSPORES* (*AMS*) [46]; and *MYB1* [47], which participates in pollen development. At S3, specifically expressed genes play key roles in the auxin response, including the orthologous gene of *TRANSPORT INHIBITOR RESPONSE 1* (*TIR1*) [48] and genes involved in the specification of floral organs, including orthologues of *AP3* and *PI* [49]. At S4, many genes, namely, *CgMYB21*, *CgMYB24*, and *CgMYB2*, are induced by jasmonate and involved in the development of stamens [50]; additionally, orthologues of *SHINE* (*SHN*) regulate floral organs through the elongation of epidermal cells [51]. At S5, there are orthologues of *ELONGATED HYPOCOTYL 5* (*HY5*), and *PHYTOCHROME A SIGNAL TRANSDUCTION 1* (*PAT1*) that play key roles in plant flowering [52,53]; orthologues of *MYB4*, *ANTHOCYANINLESS 2* (*ANL2*), and *MYB113* that play key roles in the accumulation of anthocyanins [54,55,56]; orthologues of *BEL1-LIKE HOMEODOMAIN 1* (*BLH1*) and *KNOTTED1-LIKE HOMEOBOX GENE 3* (*KNAT3*) that play key roles in ovule formation [57]; and orthologues of *CIRCADIAN CLOCK ASSOCIATED 1* (*CCA1*) involved in circadian clock regulation [58]. In summary, based on the selected GO enrichment categories and representative gene expression, flower formation occurred mainly in stage S1; the development of floral organs, including anthers, occurred in stages S1–S4; and anthocyanin biosynthesis occurred mainly in stages S4 and S5.

### 2.5. Representative Key Genes Involved in the Main Regulatory Programmes of Flower Development

According to the specifically expressed genes and their enriched GO terms, four regulatory programs, namely, flower formation, flower development, anthocyanin synthesis related to flower color, and anther development, are key functional programs of flower development in *C. giganteum*. To identify the key genes related to flower development, we screened the key candidate genes involved in the four regulatory programs mentioned above against the differentially expressed genes (DEGs) of *C. giganteum* (Appendix A) and obtained some orthologues of key genes (Appendix A and Figure 5, Appendix A). A total of 238 homologous genes related to the regulatory program of flower formation and 33, 16, 20, and 8 homologous genes related to flower development, anthocyanin synthesis, anther development, and nectary development, respectively, were found in *C. giganteum*.

Among the 238 homologous genes involved in flower formation (Appendix A), floral integrator genes, including *SUPPRESSOR OF OVEREXPRESSION OF CONSTANS 1* (*SOC1*), as well as genes involved in the flower initiation process, such as *FLOWERING TIME CONTROL PROTEIN* (*FPA*), *FLOWERING LOCUS Y* (*FY*), *PROTEIN ARGININE METHYLTRANSFERASE 5* (*PRMT5*) and *SUPPRESSOR of FRI 4* (*SUF4*), were significantly more highly expressed in the S1 and S2 stages than in the later stages. This finding is consistent with the fact that the process of floral initiation occurs in the early stage of flower bud differentiation. The two orthologues of *GA INSENSITIVE DWARF 1C* (*GID1C*), *GID1C-like1* and *CgGID1C-like2*, were highly expressed at S1 and S5. *GIBBERELLIC ACID INSENSITIVE* (*GAI*), a GA repressor, was expressed at high levels in S2–S4 and at low level in S1. *REPRESSOR OF GA 1* (*RGA1*), *REPRESSOR OF GA 2* (*RGA2*), and *REPRESSOR OF GA 3* (*RGA3*) function redundantly to repress GA-induced floral initiation. *CgRGA1*, *CgRGA2-like1*, *CgRGA2-like2*, *CgRGA2-like3*, *CgRGA2-like4*, *CgRGA2-like5*, and *CgRGA3* were expressed at lower levels in S1 but were highly redundantly expressed at S2–S5. The expression of the key GA signaling genes mentioned above indicates GA was involved in the process that occurs at S1 stage.

According to the development of flowers, the homologous genes *HOMEOBOX 51* (*HB51*), *MYB DOMAIN PROTEIN 17* (*MYB17*), *STM* and *BEL1-LIKE HOMEODOMAIN 8* (*BLH8*) play roles in maintaining the meristem or in the transformation from the vegetative to the flowering meristem. Some homologous genes were included in the ABCDE flowering model. The main A-type gene *CgAP1a*/*CgAP1b*/*CgAP1c* and its upstream gene *BLADE ON PETIOLE 2* (*BOP2*) were specifically expressed in S1, and *AP1* functioned in the determination of floral meristem and sepal identity. The type B gene of *CgPI* was expressed in S1 and S3 and has the function of specifying tepal and stamen identities. The type C gene *CgAG* functions in carpel and stamen identity and represses *CgWUS* transcription to control stem cell activity in the determination of the floral meristem [59] and was specifically expressed in S1. The four genes *CgAP2*, *LEUNIG* (*LUG*), *STERILE APETALA* (*SAP*), and *SEUSS* (*SEU*), which function in floral organ identity, negatively regulate *CgAG* [60,61]. The latter three genes were expressed preferentially in S1 and S2. The E-type genes *SEP2* and *SEP3* are involved in flower and ovule development and were redundantly expressed in S1–S5. *HUA ENHANCER 4* (*HEN4*) acts redundantly with *ENHANCER OF AG-4 1* (*HUA1*) in the specification of floral organ identity in the third whorl [62]; it was highly expressed in S1–S4. We further analyzed several representative genes using q-PCR (Figure 5C). The results showed that the expression of the *CgBLH8* gene decreased significantly after the S1 stage and remained at a low level until the S5 stage. The gene expression of *PERIANTHIA* (*PAN*) remained unchanged at S2 compared to S1, decreased to the lowest level at S4, and then increased at S5. The gene expression patterns of *WUS* and *PETAL LOSS* (*PTL*) were similar. The results for *SQUAMOSA PROMOTER BINDING PROTEIN-LIKE 8* (*SPL8*) and *INNER NO OUTER* (*INO*) were also similar. The gene expression data obtained by q-PCR were consistent with the RNA-seq data (Figure 5B).

A total of 16 homologous genes were involved in anthocyanin synthesis (Figure 5). Among them, three *PHE AMMONIA LYASE* (*PAL*) homologous genes, namely, *CgPAL1*, *CgPAL2*, and *CgPAL3*, were involved in the biosynthesis of phenylpropanoids; two homologous genes of *UDP GLUCOSE*, namely, *FLAVONOID-3-O-GLUCOSYLTRANSFERASE-1* (*UFGT1*) and *CgUFGT2*, were involved in anthocyanin biosynthesis; and all 11 other genes play key roles in the biosynthesis of flavonoids. The gene expression results showed that *CgCHS3* and *CgF3H* were preferentially expressed in S3; *CHITINASE* (*CHI*), *FLAVANONE 3′-HYDROXYLASE-2* (*F3*′*H2*) and *CgUFGT1* were preferentially expressed in S4; and the other mentioned genes, namely, *CgPAL1*, *CgPAL2*, *CgPAL3*, *CgCHS1*, *CgCHS2*, *CgF3′H2*, *FLAVONOL SYNTHASE* (*FLS*), and *CgUFGT2*, were preferentially expressed in S5. The gene expression of *CgCHI* increased considerably at S3 and S4 and then decreased at S5. The expression of the *CgANS* gene increased sharply after S2, reaching a peak level 2357 times greater than that of S1 at S4. The gene expression pattern of *CgUFGT2* is similar to that of *CgANS*. The expression level of these genes detected by q-PCR suggested that the biosynthesis of flavonoids occurs mainly during the late development period.

For anther development, some homologous genes play key roles in anther cell specification, while others are involved in mature pollen formation or anther maturation and pollen release. The *CLAVATA 1* (*CLV1*) gene and the *CgAG* and *CgAP3* genes function in maintaining floral meristem identity and specifying stamen identity, respectively [63,64,65]; all of these genes are expressed preferentially in S1. The homologous genes *BETA-AMYLASE 1* (*BAM1*), *NAB*, *ENDONUCLEOLYTIC MITOCHONDRIAL STABILITY FACTOR 1* (*EMS1*), *SOMATIC EMBRYOGENESIS RECEPTOR-LIKE KINASE 1* (*SERK1*), and *TAPETUM DETERMINANT 1* (*TPD1*) are related to anther cell specification, and most of these genes were expressed preferentially in S2 or S3. *DYSFUNCTIONAL TAPETUM 1* (*DYT1*) plays a role downstream of *EMS1*, mainly by regulating the expression of downstream genes such as *AMS*, *MALE STERILITY 1* (*MS1*) and other tapetum-preferential genes for pollen development, primarily via *TAPETAL DEVELOPMENT AND FUNCTION 1* (*TDF1*) [46]. The *CgDYT1*, *CgMYB33*, *CgTDF1*, *CgAMS*, *CgMS1*, *LESS ADHERENT POLLEN 3* (*LAP3*), and *CgMS2* genes were reported to be involved in mature pollen formation, and most of these genes were expressed preferentially in S2. The *CgMYB26* gene and the *NAC SECONDARY WALL THICKENING PROMOTING FACTOR 1* (*NST1*) gene play a role in anther maturation and pollen release. *CgMYB26* was highly expressed in S4 and S5, which was consistent with the q-PCR data (Figure 5C).

The nectary plays key roles in plant–animal interactions [66]. *YABBY5* (*YAB5*), an orthologue of an abaxial gene, was expressed throughout the whole development process in the lower part of the inflorescence and at the S1–S3 stages in the middle and upper parts of inflorescence. Liao et al. reported that *YAB5* is indispensable for pseudonectary development in *Nigella damascena* [67]. We speculated that *CgYAB5* specifies the development of the lateral organ of the nectary. Other orthologues involved in nectary development, including *BLADE ON PETIOLE 1* (*BOP1*) [68], *LATERAL ROOT PRIMORDIUM* (*LRP*) [67], *NTC-RELATED PROTEIN 1* (*NTR1*) [69], and *AGAMOUS-LIKE 20* (*AGL20*)/*SOC1*, were expressed mainly in the early stages of inflorescence development. Three orthologues of *SWEET9*, *NECTARIN 3* (*NEC3*), and *CYTOCHROME P450*, *FAMILY* 86, *SUBFAMILY* B, *and POLYPEPTIDE 1* (*CYP86B1*) are involved in sucrose export, a type of protein synthesis [70], and very long chain omega-hydroxyacid and alpha, omega-dicarboxylic acid synthesis [71], respectively. These genes are involved in nectar secretion by the nectary.

### 2.6. Representative Candidate Genes Involved in the Synchronization of Flower Growth in Different Parts of the Inflorescence

During the development of inflorescences and flowers, although the floret primordia in the lower part of the inflorescence differentiated first, they were significantly smaller than those in the middle and upper parts of the inflorescence (*p* < 0.05) at the end of stage S1. In the following stages, S2, S3, and S4, the size of the lower buds remained smaller than that of the middle and upper buds (*p* < 0.05); in particular, in stage S3, the middle buds were also significantly smaller than the upper buds. However, when the flowers bloomed in the S5 stage, there was no significant difference in the sizes of the flowers in the lower, middle, and upper parts (*p* > 0.05; Figure 6A).

To determine the reasons for the synchronous flowering of flowers in different parts of the inflorescence in *C. giganteum*, we further analyzed the key genes related to inflorescence and flower development and found that the expression patterns of some genes in different parts of inflorescence were consistent with the morphological characteristics of the inflorescence and flowers in the corresponding inflorescence parts. We found that the expression of several orthologous genes related to flower development in the lower inflorescence was delayed compared with that in the upper and middle inflorescence (Figure 6B,C). *CgWUS*, a gene that has functions in anther development and maintaining stem cells in an undifferentiated state [72], was expressed in S2–S4 in all three parts of the inflorescence. In S5, it was nearly not expressed in the upper part, was more highly expressed in the middle part, and was most highly expressed in the lower part. It is suggested that the activity of stem cells in the upper part was inactivated earlier than that in the middle part, and that in the middle part was inactivated earlier than that in the lower part. For *FRUCTOSE-BISPHOSPHATE ALDOLASE 2* (*FBA2*), *CgMYC2*, and *WRINKLED 1* (*WRI*), three genes involved in the flower formation program, the gene expression patterns were similar to those of the *CgWUS* gene. Specifically, the expression of these three genes stopped earlier in the upper part of the inflorescence than in the middle and lower parts of the inflorescence. *CgPAN*, *CgPTL*, and *CgINO*, three genes involved in flower development [73,74,75], were expressed similarly to the genes mentioned above. We also found that the expression patterns of the other orthologues were similar to that of the genes mentioned above; for example, *CgANS* and *CgMYB26*, two orthologues that participate in anther development, and *CgCHI* and *CgUFGT2*, two orthologues related to anthocyanin synthesis. The q-PCR data (Figure 6C) of the *CgFBA2*, *CgMYC2*, *CgINO*, *CgANS*, *CgMYB26*, and *CgCHI* genes were consistent with the RNA-seq data. *CgGA20OX8* was reported to negatively regulate GA activity. Therefore, the low expression in the S1 and S2 stages indicated high GA activity in these two developmental stages, which further led to flower formation. In S2, the lower level of GA activity in the lower flowers than that in the middle and upper flowers might be one of the reasons for the delay of the lower flowers.

## 3. Discussion

### 3.1. Morphological Data and GO Analysis Demonstrated the Inflorescence Development Process of C. giganteum

The transition from the vegetative growth stage to the reproductive growth stage is the key process of plant development. In some plants, vegetative growth and reproductive growth occur simultaneously, while in *C. giganteum*, inflorescence differentiation occurs in the dormant part of the underground part, similar to what occurs in many bulb plants, such as *Lilium* and *Crocus* [76,77]. Morphological observation of inflorescence and flower development is the premise for studying the molecular mechanism of flower development. In this study of *C*. *giganteum*, we found that the floral primordia differentiated during S1, and the interval between the appearance of the primordium from the first flower to the last flower was approximately three months; thus, we named this stage the floral primordium formation stage. The enriched GO terms “inflorescence development”, “maintenance of inflorescence meristem identity”, and “vegetative to reproductive phase transition of meristem” at the S1 stage also confirmed that floral primordia formed during this period. In the next stage (S2), the floral primordia developed into flower buds, and the formation of the perianth was observed first, followed by the differentiation of the pistil and stamen. However, in the early stage of pistil and stamen differentiation, neither filaments nor pollen were observed, and filaments appeared at S3. The GO terms “stamen filament development”, “jasmonic acid biosynthetic process”, “pollen sperm cell differentiation”, “another development”, and “flower development” were also enriched at the S3 stage. In the S4 and S5 stages, anther development was completed, and after blooming, anther cracking and pollen shedding indicate that anther development occurs later than pistil and stamen differentiation. The enriched GO terms “pollen sperm cell differentiation” and “double fertilization forming a zygote and endosperm” also confirmed that anther maturation occurred in the S5 stage. Therefore, according to the phenotypic observation and GO enrichment analysis of transcriptome data related to flower bud differentiation, the process of flower development after flower induction can be divided into four stages: flower primordia formation, perianth formation, pistil and stamen formation, and anther formation. This process is similar to the process of floral bub differentiation in *Lilium longiflorum* [78] and *Lilium formolongi* ‘Raizan’ [79].

In addition to inflorescence and flower organ development, flower color changes during development were also observed. We found that the markings or color of the flowers of *C*. *giganteum* appeared in the S3 stage, color deepened and formed mainly in the S4 stage, and beautiful colors and markings presented when blooming formed in the S5 stage. The enriched GO terms “terpenoid biosynthetic process” in the S3 stage and “flavonoid biosynthetic process”, “terpenoid biosynthetic process”, “positive regulation of anthocyanin metabolic process”, and “monoterpene biosynthetic process” in the S5 stage also confirmed the morphological data. This finding is consistent with the findings of Yuan et al. [31], in which the anthocyanin metabolic pathway began to be activated in the late stage of petal development in *Nigella orientalis*.

### 3.2. Autonomous Promotion and GA Promotion Together with Vernalization Promotion Might Constitute the Flower Formation Pathway of C. giganteum

The process of flower formation is regulated by a variety of pathways; among them, *FPA* and *FY* are autonomous flowering pathways [80]. The autonomous flowering pathway of *Arabidopsis* restricts the expression of the flowering suppressor *FLC*. *FPA* was found to promote flowering by inhibiting the accumulation of *FLC* mRNA [81]. The main function of *FY* is to promote polyadenylation of *FLC* to control flowering time [82]. The transcriptomic data and q-PCR data further confirmed the aforementioned development process of the inflorescence and flower of *C*. *giganteum*. The expression patterns of *CgFPA* and *CgFY*, that is, their high expression levels in the early stage of flower bud differentiation and gradual decrease during flower development, indicated that these two genes might be key genes involved in flower formation in *C*. *giganteum*. They interact with *FLOWERING CONTROL LOCUS A* (*FCA*) to inhibit *FLC* activity and can also delay flowering by activating *FLC* expression [83]. In the present study, the expression level of *CgFY* was significantly greater in the early stage of floral transformation than in the other stages of floral development in *C*. *giganteum*, which indicated that *CgFY* was more likely to play a positive role in the process of floral initiation [84].

The vernalization pathway was suggested to cooperate with the autonomic pathway to regulate the expression of *FLC* [85]. *PROTEIN ARGININE METHYLTRANSFERASE 5* (*PRMT5*) was reported to encode a type II protein arginine methyltransferase that is required for epigenetic silencing of *FLC* and for the vernalization-mediated histone modifications characteristic of the vernalized state in *Arabidopsis* [86]. Hu et al. [87] reported that FRI directly interacts with BTB proteins as well as the CUL3A ubiquitin-E3 ligase, leading to proteasomal degradation of FRI during vernalization. As two negative transcripts of *CgFLC*, the high expression of *CgPRMT5* and *CgFRI* in stage S1 decreased at S2, and nearly no expression was detected in later stages, demonstrating that the vernalization pathway is involved in flower formation in *C*. *giganteum*. Moreover, vernalization for a whole winter in an open field is necessary for flowering in *C*. *giganteum*. Taken together, these findings indicate that the autonomic pathway and the vernalization pathway constitute the flower formation pathway in *C*. *giganteum*.

The GA signals affect flower formation via the DELLA protein. The DELLA protein GAI is a repressor of GA responses and is involved in gibberellic acid-mediated flower formation [88]. The high expression level at S2–S4 and low expression level at S1 of *CgGAI* suggested that GA might function mainly at S1 and S5 during flower development in *C. giganteum*. The *GID1C* transcript encodes a GA receptor [89]. *CgGID1C-like1* and *CgGID1C-like2* were highly expressed at S1 and S5, which also indicated that GA is active at these stages. RGA1, RGA2 and RGA3 function redundantly to repress GA-induced floral initiation [90]. The low expression of *CgRGA1*, *CgRGA2-like1*, *CgRGA2-like2*, *CgRGA2-like3*, *CgRGA2-like4*, *CgRGA2-like5*, and *CgRGA3* in S1 but high redundant expression of these genes in S2–S5 demonstrated that high GA activity at the S1 stage was involved in floral initiation. Therefore, the GA pathway might also be involved in flower formation in *C. giganteum*. For the other pathways of flower formation studied in model plants, such as age and photoperiod, sporadic evidence of a few genes involved in the five flower development stages of *C. giganteum* was found. Whether these pathways are also involved in flower formation in *C. giganteum* requires further study.

### 3.3. The Varied ABCDE Flowering Model in C. giganteum

A total of eight ABCDE genes, including three class A genes, one class B gene, one class C gene, one class D gene, and two class E genes, were screened from the transcriptomic data of *C*. *giganteum* (Figure 7A). *AP1* was involved in the specific identification of two whirls of tepals [21,91]. There were three homologous genes of *CgAP1*, belonging to Class A, found in *C*. *giganteum*. We named these genes *CgAP1a*, *CgAP1b*, and *CgAP1c*, which have sequence homology to the *Lily MADS box gene 5* (*LMADS5*), *LMADS6*, and *LMADS7,* respectively [92]. In the lower flowers of *C*. *giganteum*, *CgAP1a*, *CgAP1b*, and *CgAP1c* were highly expressed, mainly in stage S1. This demonstrated that tepal identity was mainly specified at S1, the flower development period. *PI* mediates B function [24], plays a role in tepal and stamen identity, and is expressed mainly at stages S1 and S3 in *C*. *giganteum*. Pinyopich et al. reported that class C and D genes exhibit overlapping expression patterns [28]. *AG* is expressed from the early stage through the specifications of stamens and carpels [25] to the late stage through the induction of *ANTHER DEHISCENCE DEFECT 1* (*DAD1*), which catalyses the biosynthesis of JA, leading to the elongation of stamen filaments and flower opening [93]. *CgAG*, the only class C gene involved in the specific identification of stamens and carpels, is similar to the *EuAG* gene in *Eucommia ulmoides* [94] and is expressed at S1, S2, and S5 (Figure 7A). The high expression of *CgAG* at S5 might be attributed to anther splitting, pollen dispersal [95], and carpal development [96], but further research is needed. *CgSTK*, the only screened D-type homologue involved in carpel and ovule identification, was expressed mainly in stage S2 and prolonged to S4. Studies in *A. thaliana* have shown that four closely related MADS-box genes, *SEP1*, *SEP2*, *SEP3* and *SEP4*, are required to specify tepals, stamens, carpels, and ovules, respectively [26,97,98]. We screened two E-class genes in *C*. *giganteum*, namely, *CgSEP2* and *CgSEP3*, which are expressed redundantly and complementally from S1 to S5 throughout the whole development process of flowers. It is proposed that *CgSEP2* and *CgSEP3* are required for the specification of tepals, stamens, carpels, and ovules, similar to what has been observed in *A. thaliana* [26,97,98]. Based on the screened flower function genes, their gene expression in other model or non-model plants, and their expression patterns during the flower development process, we proposed an ABCDE model for *C. giganteum* (Figure 7B). A-, B- and E-class genes play key roles in outer tepals and inner tepals; B-, C- and E-class genes specify stamens; C- and E-class genes play key roles in stamen development; D-, C- and E-class genes specify carpels. It is suggested that the genes related to flower patterning in *C. giganteum* are conserved. Moreover, this model is generally similar to that of *A. thaliana* [19,20,99], and it has been shown to be suitable for *C*. *giganteum*, a species in Liliaceae [100]. However, further researched is needed to determine whether there are any new specific genes related to flower patterning in *C*. *giganteum*.

### 3.4. Preliminary Discussion on Flowering Synchronization at Different Parts of Inflorescence in C. giganteum

Floral display can be described as the number of flowers that open at one time and their arrangement in inflorescences [101], which are characteristic of flowers at the group level. The floral display of *C*. *giganteum* was different from that of lily, as the flowers of indeterminate inflorescence opened in sequence. However, the flowers of the determinate inflorescence in *C*. *giganteum* opened synchronously. Biological observation revealed that the flower primordium in the lower part of the inflorescence developed early, but the development rate decreased during the flower development period. Although the flower primordia in the middle and upper parts developed much later, they developed rapidly in the later stage and eventually reached synchronous opening. The gene expression patterns of several key genes involved in flower development, such as *CgWUS*, *CgPAN*, *CgPTL*, *CgINO*, *CgMYB26*, *CgCHI*, and *CgUFGT2*, were consistent with the biological observations mentioned above. The expression of these genes in the lower inflorescence was delayed compared with that in the upper and middle inflorescence, and the expression patterns in the upper inflorescence were similar to those in the middle inflorescence. Synchronizing the opening of a large number of flowers can increase the level of flower display, which may attract more pollinators to visit plants and increase the likelihood of mating [102]. However, synchronous flowering leads to cross-pollination of the same plant, which leads to a high self-pollination rate and a great pollen discount. There is also great risk. Once extreme weather occurs, reproduction failure occurs. The detailed mechanism and ecological value of synchronous flowering of flowers on different parts of the inflorescence in *C*. *giganteum* need further research. This will provide a theoretical reference for the synchronous flowering of ornamental plants with indeterminate inflorescence or for the simultaneous ripening of fruits of crops with indeterminate inflorescence.

## 4. Materials and Methods

### 4.1. Plant Material, Inflorescence Development Observation and Sample Collection

The plant material of *C*. *giganteum* was grown in the experimental field of the National Botanical Garden in Beijing, People’s Republic of China (116°12′34.53″ E, 40°0′5.99″ N, altitude 61.6 m). The annual average temperature and precipitation are 12.68 °C and 1163.13 mm, respectively. To investigate the developmental pattern of the inflorescence of *C*. *giganteum*, the inflorescence was artificially divided into three parts: lower, middle and upper. The morphological characteristics of the inflorescences and flowers of *C*. *giganteum* were observed from December 2020 to May 2021. According to the morphological characteristics of the inflorescences and flowers, five developmental stages, S1, S2, S3, S4, and S5, were further defined. The lengths of buds or flowers in the lower, middle and upper parts of the inflorescence at each development stage were measured, and each treatment was repeated three times. Since the inflorescence at the S1 stage was too small, only one replicate of the lower, middle and upper parts of the inflorescence was collected at this stage; three technical replicates were carried out; and three replicates of the other twelve samples were collected. All the samples were quickly frozen in liquid nitrogen and stored at −80 °C until further use.

### 4.2. De Novo Transcriptome Assembly, Differential Gene Expression Analysis, and Functional Annotation

The RNA integrity numbers (RINs) of the extracted RNA from 39 samples were greater than 9 (Appendix A). All samples were sequenced on an Illumina NovaSeq 6000 platform by Biomarker Technologies (Beijing, China). The raw reads were processed into circular consensus sequence (CCS) reads according to the adaptor. Next, full-length, non-chimeric (FLNC) transcripts were determined by searching for the polyA tail signal and the 5′ and 3′ cDNA primers in CCS. Similar full-length sequences were clustered and corrected to obtain high-quality sequences for subsequent analysis. The RNA sequencing data have been deposited in the National Center for Biotechnology Information Short Read Archive under accession number PRJNA1052943. Gene expression levels were estimated by quantifying fragments per kilobase of transcript per million fragments mapped (FPKM). Differential expression analysis of two samples was performed using edgeR. The resulting *p*-values were adjusted using Benjamini and Hochberg’s approach for controlling the false discovery rate. A *p*-value < 0.01 and a fold change ≥ 1.50 were set as the thresholds for significant differential expression. For functional annotation, the BLAST program (version 2.2.26) was used to search for associations between unigenes in various nucleotide/protein databases, namely, the NR, SwissProt, GO, COG, KOG, Pfam, and KEGG databases. Within the alignment against each database, the best aligning results were preserved.

### 4.3. Correlation Analysis between Samples

PCA, PCC, and SCC analyses were used to compare the expression profiles between samples. PCC and SCC were carried out using the cor.test function in R version 4.1.2 using the Pearson and Spearman methods, respectively. PCA was performed using the prcomp function in R version 4.1.2, using the original RPKM values as inputs.

### 4.4. Weighted Gene Co-Expression Network Analysis and Co-Expression Module Identification

Weighted gene correlation network analysis (WGCNA) [103,104] was applied to construct co-expression networks with the RPKM values of 39,985 differentially expressed genes and a coefficient of variation > 0.5 across five stages. Detailed methods are described in the work of Zhang et al. [102].

### 4.5. Identification of Specifically Expressed Genes and GO Enrichment Analysis

A specifically expressed gene was defined as being expressed at a specific developmental stage and only in this organ if its RPKM was >1.0. For the GO enrichment analysis of specifically expressed genes, protein sequences were subjected to BLAST against the non-redundant protein database of *A. thaliana*, with a cut-off value of E < 1 × 10^−10^ [105]. The GO term of each gene was subsequently assigned according to the best hit, the GO enrichment analysis of gene clusters was carried out by the agriGO program [106], and the threshold of false discovery was 0.05.

### 4.6. Expression Analysis of Genes Involved in Inflorescence and Flower Development

According to the studies of floral initiation genes, flower development genes, anthocyanin synthesis genes, anther development genes, and nectary development genes in *A. thaliana*, 238 floral initiation genes, 33 flower development genes, 16 anthocyanin synthesis genes, 20 anther development genes and 8 nectary development genes were obtained by blasting the differentially expressed genes of *C*. *giganteum* against the nucleotide/protein databases listed in Section 4.2. The expression data of these genes at different developmental stages and at different inflorescence positions were analyzed using RPKM and normalized using the Z-score method.

### 4.7. RT-qPCR Analysis

For the mRNA expression analyses, total RNA was extracted from the inflorescence, flower buds or flowers of *C*. *giganteum* using Tiangen DP762-T1C (Tiangen, Beijing, China). cDNA was synthesized using TransScript All-in-One First-Strand cDNA Synthesis SuperMIX (Transgen, Beijing, China) following the manufacturer’s protocol. q-PCR was performed using PerfectStartTM Green q-PCR SuperMix (Transgen, Beijing, China) on a Roche LightCycler^®^ 480 II PCR (Roche, Basel, Switzerland). The reaction mixture included 5 μL of 2× PerfectStartTM Green qPCR SuperMix; 0.2 μL of 10 μmol/L forward primer, 0.2 μL of 10 μmol/L reverse primer, 1 μL of cDNA, and 3.6 μL of nuclease-free H_2_O. The cDNAs were amplified following denaturation using 45 cycles of denaturation (94 °C, 30 s; 94 °C, 5 s; 60 °C, 30 s per cycle). The primers used for the q-PCR analysis are listed in Appendix A, and *CgACTIN* was used as an internal quantitative control. The relative expression values were calculated using the comparative CT (2^−ΔΔ*CT*^) method [107].

## 5. Conclusions

*C. giganteum* is distributed only in subalpine forests in East Asia, so this species is relatively unknown and related research is quite limited. This is the first study exploring inflorescence development in *C. giganteum* via transcriptomic analysis. The research results showed that the inflorescence development of *C. giganteum* has the following characteristics. First, the autonomous pathway, GA pathway, and vernalization pathway are the main flowering pathways involved. Second, in the flowering model, A + B + E determines the formation of outer and inner tepals, B + C + E determines the development of stamens, C + E determines the formation of carpels, and C + D + E determines the development of ovules. Moreover, the ABCDE model of *C. giganteum* is generally similar to that of *A. thaliana*. Interestingly, during the process of inflorescence developing into flowering, the development of florets in different parts of *C. giganteum* is “gradually synchronized”, which is rare in racemes. We speculate that the synchronous flowering in *C. giganteum* is highly important not only for plant evolution but also for plant ecology. In-depth research on the synchronous flowering of *C. giganteum* will not only help us to understand the flower development of plants with determinate inflorescences; it will also open up promising avenues for crop improvement.

## Figures and Tables

**Figure 1 plants-13-00649-f001:**
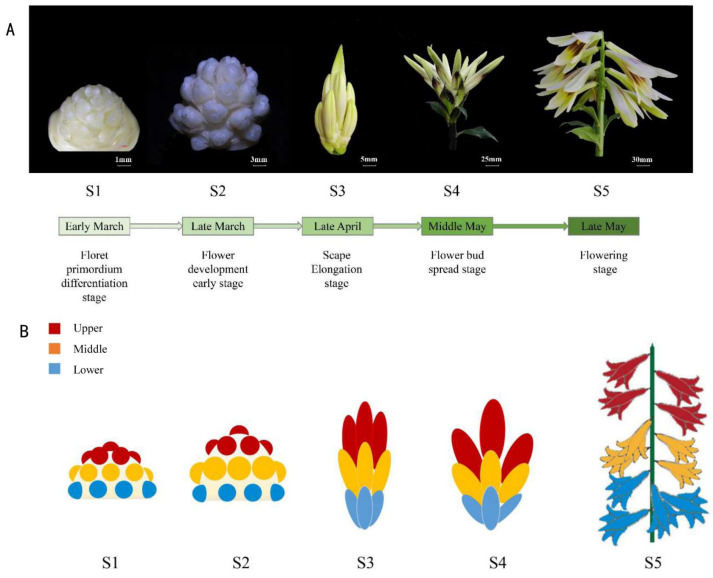
Phenology at different stages of inflorescence development in *Cardiocrinum giganteum*. (**A**) Inflorescence morphology. (**B**) Diagram of the inflorescence.

**Figure 2 plants-13-00649-f002:**
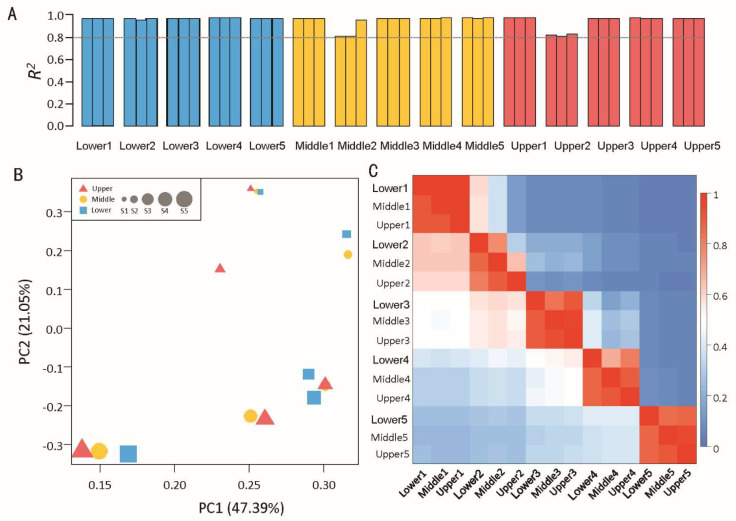
Correlation analysis of 15 samples. (**A**) Reproductive assessment of each sample in triplicate. (**B**) PCA of the 15 samples; PC1, principal component 1; PC2, principal component 2. (**C**) PCC (top right) and SCC (bottom left) analyses of the 15 samples. ‘Lower1’ indicates a sample from the lower part of the inflorescence at stage S1.

**Figure 3 plants-13-00649-f003:**
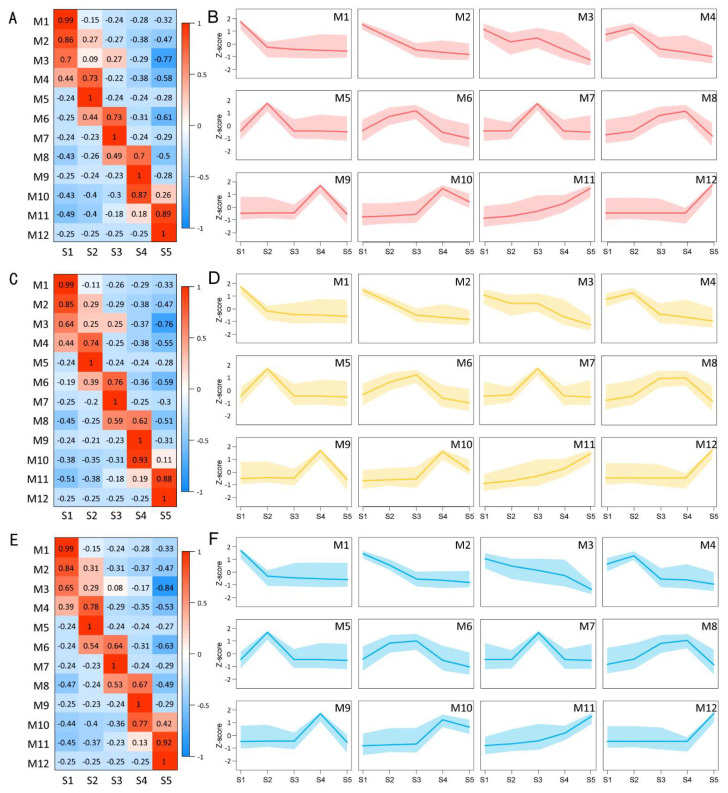
Gene co-expression modules involved in the five developmental stages of different parts of the inflorescence. (**A**) Heatmap describing the correlations between co-expression modules and developmental stages of the upper part of the inflorescence. (**B**) Expression profiles of the 12 co-expression modules in the upper part of the inflorescence. (**C**) Heatmap describing the correlations between co-expression modules and developmental stages of the middle part of the inflorescence. (**D**) Expression profiles of the 12 co-expression modules in the middle part of the inflorescence. (**E**) Heatmap describing the correlations between co-expression modules and developmental stages of the lower part of the inflorescence. (**F**) Expression profiles of the 12 co-expression modules in the lower part of the inflorescence.

**Figure 4 plants-13-00649-f004:**
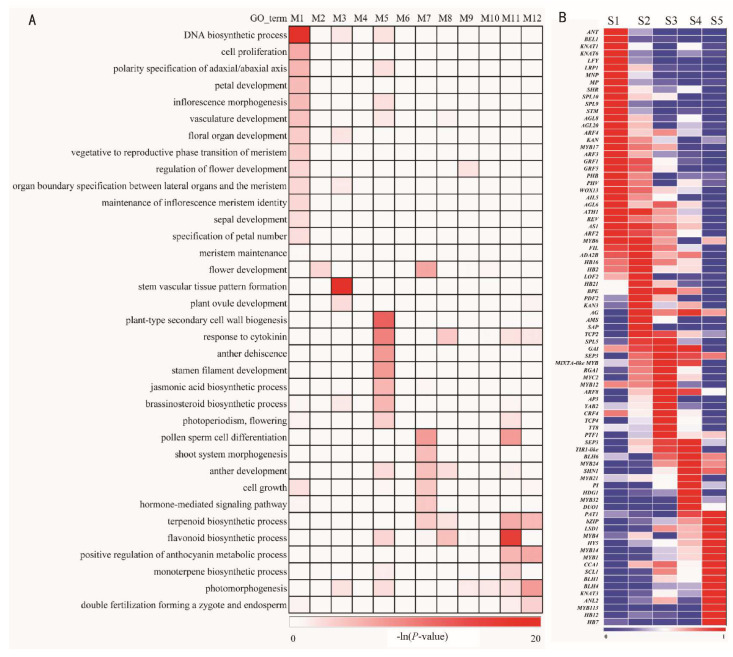
Specifically expressed genes and selected GO terms associated with inflorescence and flower development at different developmental stages in the lower part of the inflorescence. (**A**). Selected GO terms; (**B**). Specifically expressed genes. Heatmaps showing the ln-transformed GO terms (**A**) and Z-score-transformed gene expression values (**B**) at different developmental stages of the inflorescence and flower.

**Figure 5 plants-13-00649-f005:**
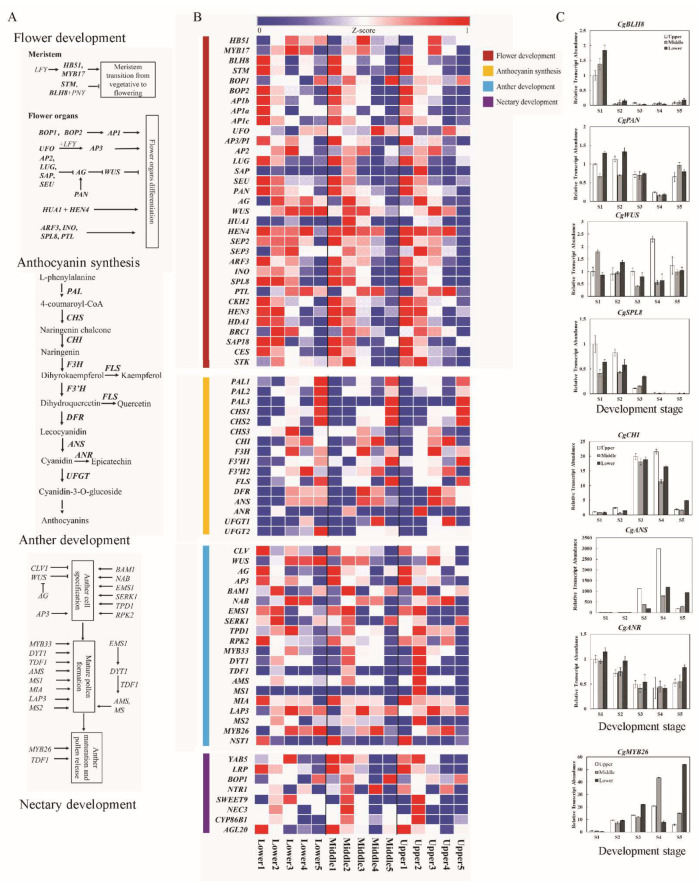
Expression of genes involved in the regulatory programs of flower development, anthocyanin synthesis, pollen development, and nectary development at different developmental stages. (**A**) Regulatory programs. Arrows indicate a promotion interaction, T-ends indicate an inhibiting genetic interaction. (**B**) Expression patterns of genes. The expression abundance of genes was normalized to the Z-score and is represented on a heatmap; the color labels to the left of the heatmap indicate that the genes were involved in different regulatory programs. (**C**) q-PCR analysis of the key representative genes. Data are expressed as the means ± SD, *n* = 3.

**Figure 6 plants-13-00649-f006:**
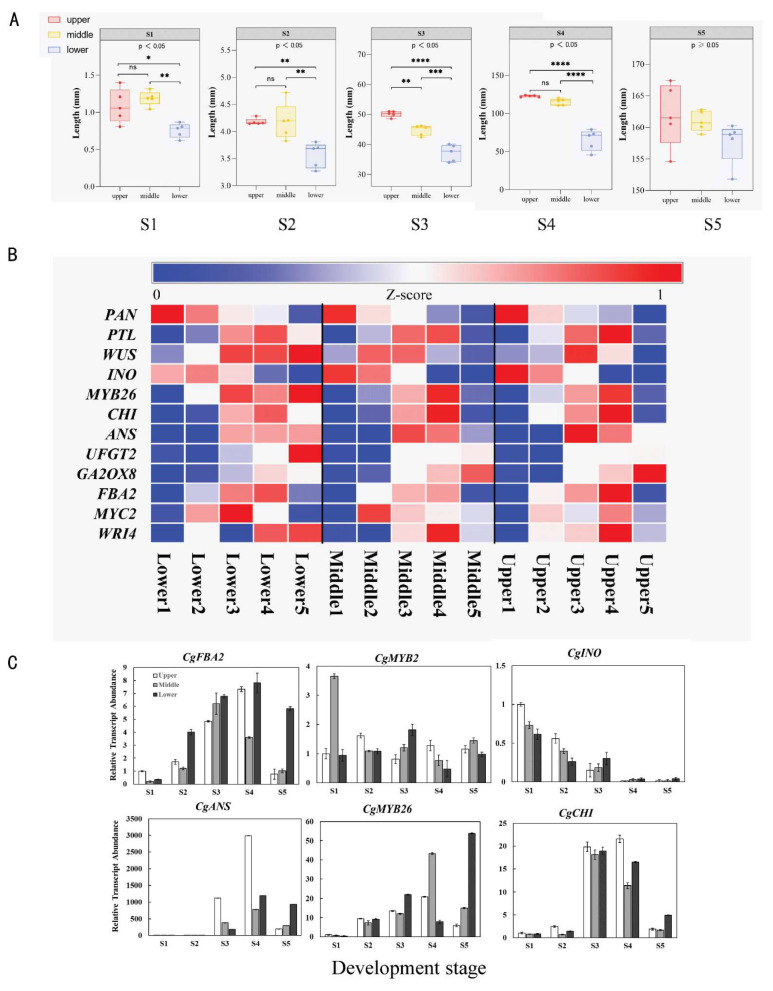
Expression profiles of the representative candidate genes involved in flowering synchronization of flower growth on different parts of the inflorescence. (**A**). Flower sizes of different parts of the inflorescence. The value bars with ns are not significant, and an asterisk denotes statistically significant differences according to a one-way analysis of variance (ANOVA), * *p* < 0.05; ** *p* < 0.01; *** *p* < 0.001, **** *p* < 0.0001. (**B**) Expression patterns of selected genes. The expression abundance of genes was normalized to the Z-score and is represented on a heatmap. (**C**) q-PCR analysis of the key representative genes. Data are expressed as means ± SD, *n* = 3.

**Figure 7 plants-13-00649-f007:**
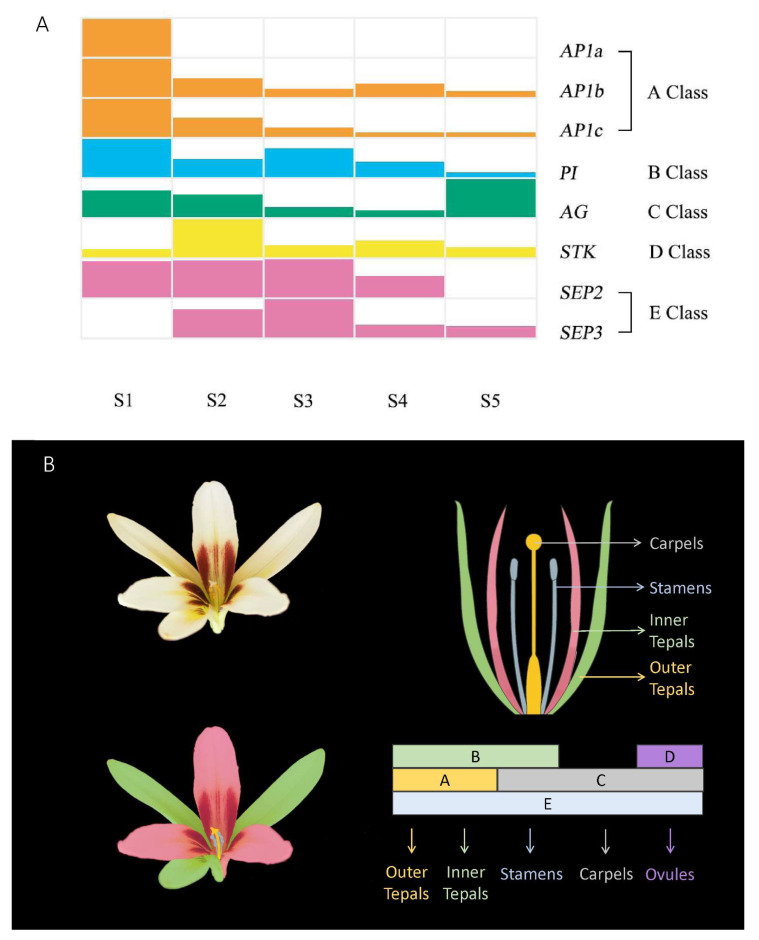
Gene expression of different functional categories (**A**) and proposed flower development model (**B**) of *C*. *giganteum*.

**Table 1 plants-13-00649-t001:** Statistics for the number of annotated transcripts.

Annotated Database	Isoform Amount
COG	17,760
GO	28,050
KEGG	21,425
KOG	28,425
Pfam	35,356
Swiss-Prot	32,322
eggNOG	43,232
NR	44,172
All	44,454

## Data Availability

Data are contained within the article or in Appendix A.

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
