# Peer review of "Time-Course Transcriptomic Analysis Reveals Molecular Insights into the Inflorescence and Flower Development of Cardiocrinum giganteum"

_plants, 2024, doi:10.3390/plants13050649_

Round 1
Reviewer 1 Report
Comments and Suggestions for Authors
The manuscript by Wei et al. employed a time-course RNA-seq analysis to dissect the genes involved in the inflorescence and flower development of Cardiocrinum giganteum. The results obtained from this work enhance our understanding of the processes and mechanisms underlying inflorescence and flower development. This information could prove valuable for regulating the flowering period and breeding of C. giganteum. The experiments were well-designed and executed, and the conclusions drawn are appropriate for the presented results. I have no concerns about the manuscript and believe it is suitable for publication.
Author Response
Thank you for your confirmation.
Reviewer 2 Report
Comments and Suggestions for Authors
The authors of the MS, attempted to dissect the pattern of inflorescence flowering in Cardiocrinum giganteum. Most of the MS authors have done well in achieving the goal
They could include the Gibberellic acid analysis between the stages, if possible to prove the stage-wise difference. Otherwise, a comparison of Gibberalic acid signaling pathway genes from RNA seq data and qRT must be shown.
Besides, in all the qRT data the authors must include standard deviation to prove their data significance
Author Response
1 They could include the Gibberellic acid analysis between the stages, if possible to prove the stage-wise difference. Otherwise, a comparison of Gibberalic acid signaling pathway genes from RNA seq data and qRT must be shown.
Reply:Thank you for your suggestion. We have analyzed Gibberellic acid biosynthesis and signaling pathway genes from RNA seq data (shown in Figure A1), and added the comparison of the genes between different stages in 2.5, 2.6 subsections of results and 3.2 subsection of discussion in the manuscript. Details are as follows:
2.5 subsection:The GA signalling genes CgGID1C-like1 and CgGID1C-like2 were highly expressed at S1 and S5. CgGAI, a GA repressor, was expressed at high levels in S2-S4 and at low level in S1. RGA1, RGA2 and RGA3 function redundantly to repress GA-induced floral initiation. CgRGA1, CgRGA2-like1, CgRGA2-like2, CgRGA2-like3, CgRGA2-like4, CgRGA2-like5, and CgRGA3 were expressed at lower levels in S1 but were highly redundantly expressed at S2-S5.
2.6 subsection:CgGA20OX8 was reported to negatively regulate GA activity. Together with the high expresssion of CgGID1C-like1 and CgGID1C-like2, the low expression of CgGAI, CgRGA1, CgRGA2-like1, CgRGA2-like2, CgRGA2-like3, CgRGA2-like4, CgRGA2-like5, and CgRGA3 in the S1 stage might indicate high GA activity in this developmental stage, which further led to flower formation.
3.2 subsection:The GA signals affect flower formation via the DELLA protein. The DELLA protein GAI is a repressor of GA responses and is involved in gibberellic acid-mediated flower formation [87]. The high expression level at S2-S4 and low expression level at S1 of CgGAI suggested that GA might function mainly at S1 and S5 during flower development in C. giganteum. The GID1C transcript encodes a GA receptor [88]. CgGID1C-like1 and CgGID1C-like2 were highly expressed at S1 and S5, which also indicated that GA is active at these stages. RGA1, RGA2 and RGA3 function redundantly to repress GA-induced floral initiation [89]. The low expression of CgRGA1, CgRGA2-like1, CgRGA2-like2, CgRGA2-like3, CgRGA2-like4, CgRGA2-like5, and CgRGA3 in S1 but high redundant expression of these genes in S2-S5 demonstrated that high GA activity at the S1 stage was involved in floral initiation. Therefore, the GA pathway might also be involved in flower formation in C. giganteum. For the other pathways of flower formation studied in model plants, such as age and photoperiod, sporadic evidence of a few genes involved in the five flower development stages of C. giganteum was found. Whether these pathways are also involved in flower formation in C. giganteum requires further study.
2 Besides, in all the qRT data the authors must include standard deviation to prove their data significance.
Reply:Thank you for your suggestion. We have added the standard deviation to all the qRT data. Details are in figure 5 and 6.

Reviewer 3 Report
Comments and Suggestions for Authors
This study delves into the inflorescence and flower development of Cardiocrinum giganteum, an East Asian species renowned for its distinctive inflorescence and medicinal bulbs. The authors employ morphological observation and time-course transcriptomic analysis to investigate the unique structure and flowering mode of C. giganteum's raceme. The findings suggest that the autonomic and vernalization pathways collectively govern flower formation in C. giganteum. The deduced flowering model reveals synchronized development across the raceme, offering insights into both evolution and ecology. The results contribute to the authors' understanding of inflorescence and flower development mechanisms, potentially aiding in the regulation of the flowering period and breeding of C. giganteum.
The work presented in this paper is commendably well-organized and well-written, providing valuable insights into the inflorescence and flower development of Cardiocrinum giganteum. The integration of morphological observation and time-course transcriptomic analysis offers a comprehensive view of the unique flowering characteristics of C. giganteum's raceme.
However, a few minor suggestions are proposed for further enhancement. In the introduction, a clearer identification of knowledge gaps and their connection to the research goals would strengthen the paper's foundation. Emphasizing the novelty and relevance of the study at the outset will engage readers and set the stage for the subsequent findings.
In the discussion section, it is recommended to place a stronger emphasis on highlighting insights gained from the findings and their applicability to future research. Addressing key questions, such as the research gaps addressed, the beneficiaries of the improvements, and potential future directions, will significantly enhance the discussion and underscore the paper's contribution.
Additionally, adding a conclusion section summarizing the key findings and their implications for future research would provide a fitting end to the paper.
Finally, it is advised to avoid using red-green color combinations in figures 2, 3, and 6 to ensure accessibility for colorblind readers.
Overall, this paper makes a valuable contribution to the field, and addressing these minor points will further enhance its clarity and impact.
Comments on the Quality of English LanguageThe overall language is commendable, demonstrating a strong grasp of the subject matter. However, there are opportunities for improvement in terms of grammar. Some sentences could benefit from restructuring to enhance the overall flow. Additionally, maintaining consistent terminology throughout the text is essential for clarity. Ambiguous phrasing should be clarified to ensure that the reader fully grasps the intended meaning. Overall, attention to these details will further elevate the quality of the manuscript.
Author Response
1 In the introduction, a clearer identification of knowledge gaps and their connection to the research goals would strengthen the paper's foundation.
Reply:Thank you for your suggestion, it will help us improve the manuscript. We have revised the introduction part, we added descriptions about knowledge gaps and their connection to the research goals. Details are as follows:
All the above-mentioned regulatory factors are based mainly on the study of model species, but the conservation, convergence, and differences in homologous genes between model and non-model species during the inflorescence and flower development are still unclear. In addition, the research on C. giganteum has focused mainly on bulb development [36], seed dormancy [37], and phylogenetic analysis [1, 38]. Only pollen development during the process of inflorescence differentiation in this species, has been studied, but the morphological and molecular mechanisms related to other developmental stages have not reported.
2 Emphasizing the novelty and relevance of the study at the outset will engage readers and set the stage for the subsequent findings.
Reply:In order to emphasize the novelty and relevance of the study at the outset, we moved the introduction of giant lily and the description of the innovation and importance of this research to the first paragraph. The detailed shown below:
Giant lily [Cardiocrinum giganteum (Wall.) Makino] is a perennial bulbous plant of the family Liliaceae, that has gorgeous lily-like flowers. This species is endemic to East Asia, and distributed mainly in temperate and subtropical woodlands in southwest China [1]. Under natural conditions, the developmental period of transition from the vegetative to the reproductive phase is long, but the flowering period is short, which limits garden application. Therefore, it is necessary to explore the flowering characteristics and underliying mechanisms of this plant to prolong its flowering period. Moreover, the racemes of C. giganteum are considered to be a transitional type from finite to infinite inflorescence and are different from those of Lilium with infinite inflorescence. C. giganteum is considered t a potentially suitable species for studying inflorescence and flower development in plants with finite inflorescences, especially the termination of the floral meristem (FM).
3 In the discussion section, it is recommended to place a stronger emphasis on highlighting insights gained from the findings and their applicability to future research. Addressing key questions, such as the research gaps addressed, the beneficiaries of the improvements, and potential future directions, will significantly enhance the discussion and underscore the paper's contribution.
Reply:Thank you for your suggestion. We have emphasized on highlighting insights gained from the findings and their applicability to future research in discussion and conclusion sections. We added some discussion on key questions, such as the beneficiaries of the improvements, and potential future directions. Details are as follows:
Sentences in discussion 3.2: The GA signals affect flower formation via the DELLA protein. The DELLA protein GAI is a repressor of GA responses and is involved in gibberellic acid-mediated flower formation [87]. The high expression level at S2-S4 and low expression level at S1 of CgGAI suggested that GA might function mainly at S1 and S5 during flower development in C. giganteum. The GID1C transcript encodes a GA receptor [88]. CgGID1C-like1 and CgGID1C-like2 were highly expressed at S1 and S5, which also indicated that GA is active at these stages. RGA1, RGA2 and RGA3 function redundantly to repress GA-induced floral initiation [89]. The low expression of CgRGA1, CgRGA2-like1, CgRGA2-like2, CgRGA2-like3, CgRGA2-like4, CgRGA2-like5, and CgRGA3 in S1 but high redundant expression of these genes in S2-S5 demonstrated that high GA activity at the S1 stage was involved in floral initiation. Therefore, the GA pathway might also be involved in flower formation in C. giganteum. For the other pathways of flower formation studied in model plants, such as age and photoperiod, sporadic evidence of a few genes involved in the five flower development stages of C. giganteum was found. Whether these pathways are also involved in flower formation in C. giganteum requires further study.
Sentences in discussion 3.3: Moreover, this model is generally similar to that of Arabidopsis thaliana [18, 19, 98], and it has been shown to be suitable for C. giganteum, a species in Liliaceae [99]. However, further researched is needed to determine whether there are any new specific genes related to flower patterning in C. giganteum.
Sentences in discussion 3.4: This will provide a theoretical reference for the synchronous flowering of ornamental plants with infinite inflorescence or for the simultaneous ripening of fruits from crops with infinite inflorescence.
Sentences in conclusion: Interestingly, during the process of inflorescence development to flowering, the development of florets in different parts of C. giganteum is "gradually synchronized", which is rare in racemes. We speculate that synchronous flowering is a transitional inflorescence type from a determinate inflorescence to an indeterminate inflorescence and is highly important not only for plant evolution but also for plant ecology. In-depth research on the synchronous flowering of C. giganteum will not only help us to understand the flower development of plants with determinate inflorescences but also open up promising avenues for crop improvement.
4 Additionally, adding a conclusion section summarizing the key findings and their implications for future research would provide a fitting end to the paper.
Reply:Thank you for your suggestion. We have added the conclusion section followed the discussion section. Details are as follows:
- Conclusions
- giganteum is distributed only in subalpine forests in East Asia, so this species is relatively unknown and related research is quite limited. This is the first study exploring inflorescence development in C. giganteum via transcriptomic analysis. The research results showed that the inflorescence development of C. giganteum has the following characteristics. First, the autonomous pathway and vernalization pathway are the main flowering pathways involved. Second, in the flowering model, A+B+E determines the formation of outer and inner petals, B+C+E determines the development of stamens, C+E determines the formation of carpels, and C+D+E determines the development of ovules. Moreover, the ABCDE model of C. giganteum is generally similar to that of A. thaliana. Interestingly, during the process of inflorescence development to flowering, the development of florets in different parts of C. giganteum is "gradually synchronized", which is rare in racemes. We speculate that synchronous flowering is a transitional inflorescence type from a determinate inflorescence to an indeterminate inflorescence and is highly important not only for plant evolution but also for plant ecology. In-depth research on the synchronous flowering of C. giganteum will not only help us to understand the flower development of plants with determinate inflorescences but also open up promising avenues for crop improvement.
5 Finally, it is advised to avoid using red-green color combinations in figures 2, 3, and 6 to ensure accessibility for colorblind readers.
Reply:Thank you for your suggestion. We have changed the red-green color combinations to red-blue color combinations in figures 2, 3, and 6. The detailed please found in manuscript.
6 The overall language is commendable, demonstrating a strong grasp of the subject matter. However, there are opportunities for improvement in terms of grammar. Some sentences could benefit from restructuring to enhance the overall flow. Additionally, maintaining consistent terminology throughout the text is essential for clarity. Ambiguous phrasing should be clarified to ensure that the reader fully grasps the intended meaning. Overall, attention to these details will further elevate the quality of the manuscript.
Reply:Thank you for your suggestion. We have worked on both language and readability and have also involved native English speakers for language corrections. The detailed please found in the manuscript.

Reviewer 4 Report
Comments and Suggestions for Authors
The authors did RNA seq analysis of the flowering stages of Cardiocrinum giganteum.
I find most of the analysis satisfying and properly written some minor comments
Write citations and details of methods, and write clearly cutoffs and details so anyone else can verify your result by following the same parameters.
Write details of RIN number and quality of RNA used, also, provide a table count of aligned genes and details of which database was used for the alignment.
Author Response
1 Write citations and details of methods, and write clearly cutoffs and details so anyone else can verify your result by following the same parameters.
Reply:Thank you for your suggestion. We have added citations and details of methods. The detailed please found in section 4 of the manuscript.
- Write details of RIN number and quality of RNA used, also, provide a table count of aligned genes and details of which database was used for the alignment.
Reply:Thank you for your suggestions. We have added the RIN number and quality of RNA used to Table A1 and modified the related description in subsection 4.1 and 4.2 of method and material section of the manuscript. The aligned genes and details of which database was used for the alignment, were listed in Table A4. Details of the related description of RNA quality are as follows:
4.1 subsection:Since the inflorescence at S1 stage was too small, only one replicate of the lower, middle and upper part of inflorescence at this stage was collected, three technical replicates were carried out; three replicates of other twelve samples were collected. All samples were quickly frozen in liquid nitrogen and stored at -80℃ until further required.
4.2 subsection:The RNA integrity numbers (RINs) of the extracted RNA from 39 samples were greater than 9 (Table A1).
